# CAST: CLUSTERING SELF-ATTENTION USING SURROGATE TOKENS FOR EFFICIENT TRANSFORMERS

## ABSTRACT

The Transformer architecture has shown to be a powerful tool for a wide range of tasks. It is based on the self-attention mechanism, which is an inherently computationally expensive operation with quadratic computational complexity: memory usage and compute time increase quadratically with the length of the input sequences, thus limiting the application of Transformers. In this work, we propose a novel Clustering self-Attention mechanism using Surrogate Tokens (CAST), to optimize the attention computation and achieve efficient transformers. CAST utilizes learnable surrogate tokens to construct a cluster affinity matrix, used to cluster the input sequence and generate novel cluster summaries. The self-attention from within each cluster is then combined with the cluster summaries of other clusters, enabling information flow across the entire input sequence. CAST improves efficiency by reducing the complexity from $O(N^2)$ to $O(\alpha N)$ where $N$ is the sequence length, and $\alpha$ is constant according to the number of clusters and samples per cluster. We show that CAST performs better than or comparable to the baseline Transformers on long-range sequence modeling tasks, while also achieving higher results on time and memory efficiency than other efficient transformers.

## 1 INTRODUCTION

The Transformer architecture (Vaswani et al., 2017) has revolutionized many fields within machine learning such as translation (Vaswani et al., 2017), summarization (Miller, 2019), text generation (Chen et al., 2019), sentiment classification (Sun et al., 2019), and also tasks like image classification (Dosovitskiy et al., 2020), object detection (Liu et al., 2021b), and protein folding (Jumper et al., 2021). The self-attention mechanism stands at the core of its strengths. It allows the Transformer to directly model long-range dependencies within a sequence without the need for a hidden state like in recurrent neural networks (Hochreiter & Schmidhuber, 1997). However, the self-attention mechanism has an inherent large memory cost, since its complexity grows quadratically with the input sequence length. With these memory requirements and the ever-increasing size of large language models, such as the GPT series (Brown et al., 2020; OpenAI, 2023) and LLaMA (Touvron et al., 2023), a need for more efficient attention mechanisms has emerged (Dao et al., 2022). Current implementations of more efficient self-attention mechanisms can be roughly grouped into the following categories: (1) apply self-attention on subsets of the input sequences(sparsification) (Ainslie et al., 2020; Daras et al., 2020; Kitaev et al., 2020; Ma et al., 2023; Tay et al., 2020b; Zaheer et al., 2021), (2) approximate the self-attention mechanism with a lower complexity (Choromanski et al., 2020; Liu et al., 2021a; Wang et al., 2020), and (3) remove self-attention in favor of a lower complexity similar operation (Gu et al., 2022; Lee-Thorp et al., 2021; Smith et al., 2023; Tolstikhin et al., 2021).

In this paper, we introduce a new efficient variant of self-attention, named *Clustering Attention using Surrogate Tokens* (CAST) – see Figure 1. It applies clustering to self-attention and introduces two novel ideas, namely 1) learnable clustering of tokens, and 2) cluster summaries, which allow for information to flow between tokens from different clusters within the same attention head. CAST learns to cluster tokens that would have a strong connection in the original attention matrix by clustering based on a similarity matrix between the surrogate tokens, queries, and keys. Standard self-attention is applied within clusters, where its result is combined with cluster summaries based on a previously created similarity matrix. This allows for each token to retrieve information from the rest of the sequence, improving stability and performance. CAST is significantly faster than other efficient Transformer variants, and matches or improves the performance of a standard Transformer on long-sequence modeling tasks.

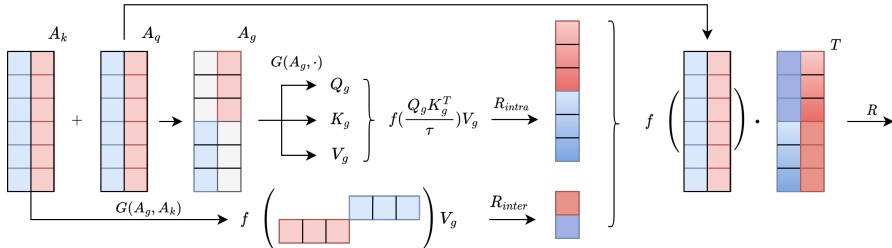

Figure 1: Sketch of the proposed method. The colors red and blue correspond to two different clusters. With the queries ($Q$), keys ($K$), and values ($V$), we create the surrogate token similarities $A_q$ (similarity between the queries and surrogate tokens) and $A_k$ (similarity between the keys and surrogate tokens). They are combined to create a final similarity $A_g$ for each token to each cluster. We then use this clustering of tokens and create the clustered queries ($Q_g$), keys ($K_g$), and values ($V_g$). Within each cluster, self-attention is applied resulting in $R_{intra}$. Furthermore, $A_k$ is also clustered and matrix multiplied with $V_g$ to create a summary per cluster resulting in $R_{inter}$. The results $R_{intra}$ and $R_{inter}$ are then combined using $A_q$ as the weights for a weighted sum, resulting in $R$. Another linear projection $O$ is then applied on $R$ and passed on to the feedforward layer of the Transformer.

The **contribution** of this paper is a novel efficient approach to replace the self-attention in Transformers based on **learning token clusters** in an unsupervised manner.

## 2   RELATED WORKS

Prior research on efficient computation of self-attention has focused on chunking attention, clustering attention, and the current state-of-the-art, structured-state-space-based models.

**Chunking attention.** One obvious way to solve the quadratic complexity of self-attention is to chunk the given sequence into smaller pieces and apply self-attention within those pieces. This is known as Local Attention (Luong et al., 2015). However, by chunking the sequence, no information can be passed between chunks, causing a decrease in task performance below that of the standard Transformer. Several works have sparsified the attention matrix by windowing or chunking the sequence (Ainslie et al., 2020; Beltagy et al., 2020; Child et al., 2019; Luong et al., 2015; Zaheer et al., 2021). Some opted for applying attention in a sliding window manner, but the use of global attention to special tokens, such as the "CLS", is also common among the original efficient Transformers of the LRA benchmark (Ainslie et al., 2020; Beltagy et al., 2020; Zaheer et al., 2021). These models also use the "CLS" token for the final classification, allowing all parts of the sequence to contribute to the final result. BigBird (Zaheer et al., 2021) combined global attention, window attention, and random blocks of attention to achieve state-of-the-art performance on the LRA benchmark. Despite the efficiency gains of the chunking of self-attention, it does not necessarily model long-range dependencies well. Multiple rounds of self-attention can be necessary to create a large enough receptive field to model long-range dependencies. Although chunking has shown its effectiveness, it does not model long-range dependencies well, since no information can flow from distant parts of the input sequence.

**Clustering attention.** One way to easily model long-range dependencies is the clustering of the input sequence which is only partially dependent on the order of the input. Specifically, the Reformer (Kitaev et al., 2020) and its descendant SMYRF (Daras et al., 2020) both use locality-sensitive hashing (LSH) to apply clustering to the input sequence and then apply a form of attention. The Reformer first uses the constraint of the queries and keys being equal such that the attention matrix is symmetric. Then to create the clusters they define a random matrix $R \in \mathbb{R}^{d_h \times N_c/2}$, which is then matrix multiplied with query-key. The query keys are then clustered based on the result of $argmax([X_{qk}R \oplus -X_{qk}R])$. The symbol $\oplus$ stands for concatenation, while $X_{qk}$ stands for the shared query-key representation. The resulting clusters are of different sizes, making it difficult to compute them on conventional hardware, harming efficiency, f.i. on the Long Range Arena Benchmark.

SMYRF efficiently computes asymmetric clustering of queries and keys, that is a query is not necessarily clustered with its corresponding key. Unlike the Reformer, SMYRF also creates balanced clusters of constant size and thus achieving better computational efficiency. Although both these

clustering Transformers do model long-range dependencies, clustering also creates problems. The random initialization of the network causes queries and keys to be clustered randomly at first. As a result, the weight update for queries and keys is based only on the information that is inside single clusters. Furthermore, gradients could be unstable when a query or key switches from one cluster to another. Ideally, an efficient Transformer retains the original strength of the Transformer, namely the information flow throughout the entire input sequence via the self-attention mechanism. Our approach keeps this information flow and introduces more stability through the use of cluster summaries, which act as an indicator of the type of information that can be obtained in a given cluster.

**Structured State Space Models.** More recent work on efficient sequence models includes Structured State Space Models (SSSM), such as S4 (Gu et al., 2022), S5 (Smith et al., 2023), and MEGA (Ma et al., 2023) which are currently the state-of-the-art on long-range sequence modeling benchmarks. SSSMs do not use the self-attention mechanism, and rely on a learnable state space to capture relevant dependencies in a sequence. However, we do not investigate these types of architectures further, as they are no longer related to the Transformer architecture and thus the self-attention mechanism.

## 3 CLUSTERING ATTENTION USING SURROGATE TOKENS

We propose CAST, Clustering Attention using Surrogate Tokens, that learns token clusters and optimizes the computation of attention in Transformer architectures. We present its versions with single-headed and multi-headed self-attention, and discuss the computational complexity. A visualization of CAST is in Figure 1, and a module-based description is in Appendix A.1. Moreover, a nomenclature with symbol definitions and pseudocodes are in Appendices A.2 and A.3.2, respectively.

### 3.1 INTUITION

A query ($Q$) and key ($K$) which are in the same direction with a large enough magnitude will end up with a large score in the self-attention matrix $A_K$. This relationship can be exploited for clustering by defining some static clustering directions, determining the similarity of all queries and all keys with these clustering directions, and then clustering based on this similarity. However, this approach has two problems: (1) when clustering, directions are randomly initialized, and their configuration might not be optimal for the task that is trained on, and (2) when training is started, queries and keys are clustered randomly. Consequentially, the gradient of the queries and keys is only based on the self-attention within their cluster, making it impossible for queries and keys from different clusters to align themselves according to the loss.

To alleviate this problem, we design CAST to ensure that the clustering directions are learnable and that each token receives information from all clusters. In CAST, surrogate tokens represent the learnable clustering directions and are used as a surrogate for finding similar queries and keys. The weight of each cluster is based on the similarity of its query and the clustering direction. Within the cluster of a certain token, we apply self-attention. For other clusters, cluster summaries are created, based on the similarity of a token key with the direction of the cluster it belongs to.

### 3.2 SINGLE-HEAD CLUSTERING ATTENTION USING SURROGATE TOKENS

CAST is an extension of the self-attention mechanism in the Transformer architecture (Vaswani et al., 2017). We first create query-key-value combinations from the input sequence $\mathbf{X} \in \mathbb{R}^{N \times d}$, where $N$ is the input sequence length, and $d$ the feature embedding dimension:

$$\mathbf{Q} = \mathbf{X}W_q, \qquad \mathbf{K} = \mathbf{X}W_k, \qquad \mathbf{V} = \mathbf{X}W_v, \qquad \in \mathbb{R}^{N \times d}, \qquad (1)$$

where $W_q, W_k, W_v \in \mathbb{R}^{d \times d}$ are learnable parameters for the queries ($\mathbf{Q}$), keys ($\mathbf{K}$), and values ($\mathbf{V}$) respectively. To create clusters and lower the computational complexity, we define learnable surrogate tokens $\mathbf{S} \in \mathbb{R}^{N_c \times d}$, where $N_c$ indicates the number of clusters. The surrogate tokens represent the learnable clustering directions and are used as a surrogate for finding similar queries and keys. Then we compute the similarity matrices with the surrogate tokens for the queries ($\mathbf{A}_q$) and the keys ($\mathbf{A}_k$). We combine these similarity matrices using a ratio $\sigma(\varphi) : 1 - \sigma(\varphi)$ based on a linear transformation of $X$, where $\sigma$ indicates the sigmoid function.

$$\mathbf{A}_q = \mathbf{Q}\mathbf{S}^T, \mathbf{A}_k = \mathbf{K}\mathbf{S}^T \qquad\qquad \in \mathbb{R}^{N \times N_c}$$

$$\varphi = \mathbf{X}W_\varphi + b_\varphi \qquad\qquad \in \mathbb{R}^{N \times 1}$$

$$\mathbf{A}_g = \sigma(\varphi) \odot f_2(\mathbf{A}_q) + (1 - \sigma(\varphi)) \odot f_2(\mathbf{A}_k) \qquad\qquad \in \mathbb{R}^{N \times N_c}, \qquad (2)$$

where $\sigma(\varphi)$ is a sigmoid function applied to a linear transformation of $\mathbf{X}$, which is represented as $\mathbf{X}W_\varphi + b_\varphi$. Where $W_\varphi \in \mathbb{R}^{d \times 1}$ and $b_\varphi \in \mathbb{R}^1$ are learnable parameters. The function $f(\cdot)$ indicates an attention function, which in this paper includes the classical softmax and the Laplace function from MEGA (Ma et al., 2023). In the case of softmax, $f_i(\cdot)$ indicates that the softmax is applied over the dimension $i$ of the matrix. Here the softmax is applied to the dimensions holding corresponding to the different clusters. The symbol $\odot$ represents element-wise multiplication. Subsequently, the calculated similarities are used to cluster the input sequence using a clustering mechanism $G$, computed as $G : \mathbb{R}^{N \times N_c}, \mathbb{R}^{N \times *} \to \mathbb{R}^{N_c \times \kappa \times *}$, where $*$ indicates any given shape, and $\kappa$ indicates the size of a cluster. Furthermore, let $G^{-1}$ indicate the reverse of the function $G$, such that $G^{-1} : \mathbb{R}^{N \times N_c}, \mathbb{R}^{N_c \times \kappa \times *} \to \mathbb{R}^{N \times *}$, where in the event of an input is contained in two clusters the sum is calculated. Then, standard self-attention is applied within each cluster as follows:

$$\mathbf{R}_{intra} = f\left(\frac{\mathbf{Q}_g \mathbf{K}_g^T}{\tau}\right)\mathbf{V}_g \qquad \in \mathbb{R}^{N_c \times \kappa \times d}, \tag{3}$$

where $\mathbf{Q}_g = G(\mathbf{A}_g, \mathbf{Q})$, $\mathbf{K}_g = G(\mathbf{A}_g, \mathbf{K})$, $\mathbf{V}_g = G(\mathbf{A}_g, \mathbf{V})$, and $\tau$ is a scalar depending on the used attention function. Here $\mathbf{R}_{intra}$ indicates the result of attention within the clusters.

To create a gradient between tokens from different clusters we apply attention between clusters as well. To do this, we define value summaries $R_{inter}$, which is a weighted sum of all values within each cluster where the weights $\mathbf{A}_{inter}$ are based on $\mathbf{A}_k$ and $\varphi$ as follows:

$$\mathbf{A}_{inter} = G\left(\mathbf{A}_g, \frac{\mathbf{A}_k \odot \phi(-\boldsymbol{\varphi})}{\tau_k}\right)\mathbf{I}'_{N_c} \qquad \in \mathbb{R}^{N_c \times \kappa \times 1},$$

$$\mathbf{R}_{inter} = f_2\left(\mathbf{A}_{inter}\right)\mathbf{V}_g^T \qquad \in \mathbb{R}^{N_c \times 1 \times d}, \tag{4}$$

where $I'_{N_c}$ indicates the expanded identity matrix $I_{N_c}$ such that $I'_{N_c} \in \mathbb{B}^{N_c \times N_c \times 1}$, the function $\phi(x) = \text{Softplus}(x) + 1$ (Zheng et al., 2015), and $\tau_k$ is a scaling factor. We use $A_Q$ and $r$ to create an attention matrix used as a weighted sum for the value summaries and attention within clusters:

$$\mathbf{A}_{sum} = f_3\left(\frac{\mathbf{A}_q \odot \phi(\boldsymbol{\varphi})}{\tau_q}\right) \qquad \in \mathbb{R}^{N \times N_c},$$

$$\mathbf{A}_{inter} = (\mathbf{A}_{sum} \odot \hat{M}) \qquad \in \mathbb{R}^{N \times N_c},$$

$$\mathbf{A}_{intra} = G(\mathbf{A}_g, \mathbf{A}_{sum} \odot M) \qquad \in \mathbb{R}^{N \times N_c},$$

$$\mathbf{R} = G^{-1}(\mathbf{A}_g, \mathbf{A}_{intra}\mathbf{R}_{intra}) + \mathbf{A}_{inter}\mathbf{R}_{inter} \qquad \in \mathbb{R}^{N \times d}, \tag{5}$$

where $M \in (0,1)^{N \times N_c}$ is a mask where $M_{i,j} = 1$ if $X_i \in G(\mathbf{A}_g, \mathbf{X})_j$ and $M_{i,j} = 0$ if $X_i \notin G(\mathbf{A}_g, \mathbf{X})_j$. As a result of this operation, $R$ is a weighted sum according to $A_{sum}$ of the attention within clusters ($\mathbf{R}_{intra}$) and the summaries of the clusters ($\mathbf{R}_{inter}$). The final output $\mathbf{O}$ is then calculated as $\mathbf{O} = \mathbf{R}W_o \in \mathbb{R}^{N \times d}$, where $W_o \in \mathbb{R}^{d \times d}$ are learnable parameters. $\mathbf{O}$ is then passed on to the rest of the standard Transformer architecture.

**Clustering.** The clustering mechanism is an integral part of CAST, and serves to group inter-important tokens of the input sequence. We measure the inter-importance as the similarity scores in the attention matrix $\mathbf{A}_g$. We define two clustering mechanisms to maximize the similarity per cluster:

A) *Top-K Clustering Mechanism.* The *Top-K* clustering mechanism is a naive approach to clustering the input sequence, the indices of the largest $K$ elements in $\mathbf{A}_g$ are taken per cluster and used to index the original sequence. Because *Top-K* simply maximizes the similarity scores per cluster separately, it is possible for any token to be contained in anywhere between 0 and $N_c$ clusters. This attribute of *Top-K* can be useful in case padding is used, by setting the similarity scores of padding to 0, it can be ensured that padding is never taken into consideration when applying attention within clusters. However, in static sequence domains, like images, it can also cause certain parts of the input sequence to never be clustered.

B) *Single Assignment Top-K Clustering Mechanism.* This approach has the constraint that every part of the sequence can only be assigned to a single cluster, and ensure that every token is part of the result of CAST and thus has a gradient. We implement the constraint by clustering tokens in descending order according to their maximum score in $\mathbf{A}_g$. When a cluster has reached the desired size, we no longer assign tokens to this cluster.

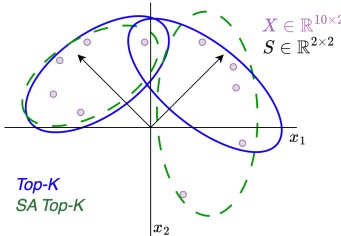

Figure 2: The practical difference between the *Top-K* and *SA Top-K* clustering mechanisms. Here, $S$ indicates the clustering direction of two surrogate tokens. The blue and green dashed circles indicate the clusters that the *Top-K* and SA *Top-K* clustering mechanisms would create, respectively.

### 3.3 Multi-Head Clustering Attention using Surrogate Tokens

To apply CAST in a multi-headed scenario, the surrogate tokens $\mathbf{S}$ are also split into multiple heads such that $S \in \mathbb{R}^{N_c \times h \times d_h}$, where $h$ is the number of heads, and $d_h = \frac{d}{h}$. The score $\mathbf{A}_g$ is then computed as follows:

$$
\begin{aligned}
\mathbf{A}_q &= \mathbf{Q}\mathbf{S}^T, \mathbf{A}_k = \mathbf{K}\mathbf{S}^T && \in \mathbb{R}^{N \times h \times N_c}, \\
\boldsymbol{\varphi} &= \mathbf{X}W_\varphi + b_\varphi && \in \mathbb{R}^{N \times 1}, \\
A_q^s &= \sigma(\boldsymbol{\varphi}) \odot f_2(\sum_h \mathbf{A}_{q_{:,h,:}}) && \in \mathbb{R}^{N \times N_c}, \\
A_k^s &= (1 - \sigma(\boldsymbol{\varphi})) \odot f_2(\sum_h \mathbf{A}_{k_{:,h,:}}) && \in \mathbb{R}^{N \times N_c}, \\
\mathbf{A}_g &= A_q^s + A_k^s && \in \mathbb{R}^{N \times N_c}.
\end{aligned}
\tag{6}
$$

In short, we sum the similarity scores $\mathbf{A}_q$ and $\mathbf{A}_k$ over the head dimension to get the similarity of each token to each cluster. After this step CAST works as described in Section 3.2, but with an added constant dimension $h$. Before the result $\mathbf{O}$ is calculated, the result of the different heads $R$ is concatenated such that $\mathbf{R} \in \mathbb{R}^{N \times d}$.

### 3.4 Complexity

With the use of CAST, the original quadratic complexity of the self-attention mechanism is significantly reduced. The complexity of CAST without added constants regarding the number of layers, batch size, and hidden dimensions is $O(\alpha N)$. Here $\alpha = \max(\kappa, N_c^2)$, where $\kappa$ is the number of elements in a cluster, and $N_c$ the number of clusters. Here, the complexity $O(N\kappa)$ is derived from the computation of $\mathbf{R}_{intra}$ being $N_c \kappa^2$, which can be rewritten as $N\kappa$. The complexity $O(NN_c^2)$ is derived from the computation of $\mathbf{R}_{inter}$. Here, we have set the relation of $\kappa$ to be $\kappa = \frac{N}{N_c}$, although technically not necessary this would allow for each token to be clustered. Theoretically, the memory usage is lowest with a configuration where $N_c^2 = \kappa$.

## 4 Experimental Setup

We use the Long Range Arena Benchmark (Tay et al., 2020c) (LRA) to evaluate the performance of CAST, and compare the results with those of other methods for efficient Transformers. The LRA benchmark is composed of six complex tasks on different data modalities and sequence length (1K-16K tokens), considered to theoretically represent various challenging tasks and complexity levels. The LRA dataset and training rules were proposed with the aim of allowing researchers to compare efficient Transformers without the need for large computational power. We further perform an ablation study on the surrogate tokens to determine how the number of clusters influences the performance, peak memory usage, and the training steps efficiency. Lastly, we analyze the learned clusters to gain insights into why CAST works.

The tasks we consider serve to investigate the capability of models to deal with a diverse range of data modalities and structures, such as natural language, images, and mathematics. The LRA benchmark is currently being used as the main benchmark for efficient Transformers and long-range sequence

modeling. The evaluation metric for all the tasks in LRA is classification accuracy. More details on the pertinence of the LRA and its six tasks can be found in Appendix A.4.

## 4.1 EXPERIMENTS

We carried out experiments on a variety of hardware, and expanded upon in more detail per experiment where significant. For reproducibility and fair comparison, we keep the number of layers and features comparable to those used in efficient Transformers in the original LRA paper (Tay et al., 2020c).

**CAST efficiency.** We evaluate the efficiency of CAST by running it on the *Text* task of LRA with a varying sequence length of 1K, 2K, 3K, and 4K. For each of these sequence tasks, we determine the peak memory usage and the number of training steps per second relative to the original Transformer architecture. For comparison with other efficient Transformers we take their performance reported in the original LRA paper (Tay et al., 2020c). We ensure that CAST and the Transformer use the exact same hyperparameters, such as the number of layers, the number of heads, and the size of the feature space. CAST uses a cluster size of 200 throughout all sequence lengths. All experiments regarding memory and time efficiency were run on a single A40 GPU.

**Long Range Arena performance.** We evaluate the performance of CAST on the LRA dataset by performing a small hyperparameter sweep. In total, we ran ten full-length training sessions per task, where the checkpoint with the lowest validation loss was used to evaluate the performance of CAST. In Appendix A.5, a more detailed description regarding the hyperparameters can be viewed. Furthermore, a Weights & Biases (Biewald, 2020) report with all hyperparameters and loss curves can be found here[1].

**Clustering ablation.** We further perform an ablation study on how the number of surrogate tokens, i.e. the number of clusters, affects the performance, peak memory usage, and number of training steps per second. We also investigate whether there is a difference in using the *Top-K* or Single Assignment *Top-K* clustering mechanisms in the *Image* task. For this ablation, we use the *Text* and *Image* tasks to determine whether there is a difference between modalities. For each task, we take the best-performing models from the hyperparameter sweep but vary the cluster size $\kappa$ such that $\kappa \in \{32, 64, 128, 256, 512\}$.

**Visual analysis on clusters.** Lastly, we perform a visual analysis on the learned clusters in the *Image* task of the LRA dataset. From the ablations, we take a single model with two CAST layers and eight surrogate tokens. We then visualize which tokens are clustered together and have a more in-depth look at the obtained similarity scores $\mathbf{A}_g$.

## 5 RESULTS AND DISCUSSION

### 5.1 LONG RANGE ARENA EFFICIENCY

In Table 1, we compare the speed and memory efficiency of several notable architectures. We observe that CAST with *Top-K* is significantly faster compared to both the original Transformer and other efficient Transformers for all sequence lengths, with it being 6.18 times faster during training than the original Transformer on a sequence length of 4K. Furthermore, CAST needs slightly less memory than other efficient Transformers, only needing 10% of the memory compared to the original Transformer architecture at a sequence length of 4K. The use of *SA Top-K* lowers the speed of CAST significantly but does not affect the memory efficiency. Further results regarding the efficiency during inference can be found in Appendix A.6.1.

### 5.2 LONG RANGE ARENA PERFORMANCE

Table 2 reports the performance results of CAST compared to those of the baseline Transformer and its efficient variations, and the current state-of-the-art models. CAST achieves performance between that of the state space models and the other efficient Transformers. Although structured state space models are state-of-the-art, they cannot be directly compared to other efficient Transformers since they apply global convolutions and are not solely relying on attention. CAST has a relatively high score for the *Image* task and a relatively low score for the *Pathfinder* task compared to that of the other efficient Transformers. The low score of the *Pathfinder* task could be explained by the fact that many of the pixels in the Pathfinder image are black, which makes their query-key pairs similar and put in the same cluster. An extended version of Table 2 can be found in Appendix A.6.2.

---

[1]Link to be publicly available upon acceptance

Table 1: Speed and Memory efficiency of the LRA Benchmark with the average performance (Avg.). The Transformer and CAST were created using the same hyperparameters. A batch size of 25 was used and CAST uses a constant cluster size of 200. Speed and Memory increase/decrease are reported relative to the results of the original Transformer architecture. Models annotated with the † symbol had their relative speed and memory taken from the LRA benchmark (Tay et al., 2020c).

| Model | Steps Per Second ↑ | | | | Peak Memory Usage ↓ | | | | Avg. Performance |
|---|---|---|---|---|---|---|---|---|---|
| | 1K | 2K | 3K | 4K | 1K | 2K | 3K | 4K | |
| Transformer (Vaswani et al., 2017) | 1.0 | 1.0 | 1.0 | 1.0 | 1.0 | 1.0 | 1.0 | 1.0 | 57.71 |
| Reformer† (Kitaev et al., 2020) | 0.5 | 0.4 | 0.7 | 0.8 | 0.56 | 0.37 | 0.28 | 0.24 | 50.56 |
| Sinkhorn Trans.† (Tay et al., 2020b) | 1.1 | 1.6 | 2.9 | 3.8 | 0.55 | 0.31 | 0.21 | 0.16 | 51.23 |
| Performer† (Choromanski et al., 2020) | 1.2 | 1.9 | 3.8 | 5.7 | 0.44 | 0.22 | 0.15 | 0.11 | 51.18 |
| Luna-16 (Ma et al., 2021) | 1.2 | 1.8 | 3.7 | 5.5 | 0.44 | 0.23 | 0.17 | **0.10** | 59.55 |
| S4 (Gu et al., 2022) | - | - | - | 4.8 | - | - | - | 0.14 | 86.09 |
| MEGA (Ma et al., 2023) | - | - | - | 2.9 | - | - | - | 0.31 | 88.21 |
| MEGA-Chunk (Ma et al., 2023) | - | - | - | 5.5 | - | - | - | 0.13 | 85.66 |
| CAST (Top-K) | **1.76** | **3.25** | **4.48** | **6.18** | **0.33** | **0.18** | **0.13** | **0.10** | 59.32 |
| CAST (SA Top-K) | 1.47 | 2.24 | 2.33 | 2.62 | **0.33** | **0.18** | **0.13** | **0.10** | 57.57 |

.

Table 2: The performance of different architectures on the Long Range Arena benchmark in classification accuracy. We divide these works in (A) Transformer architectures that do not use Structured State Spaces or any derivation of this, and (B) Architectures using Structured State Spaces. (A-Top) The original Transformer architecture. (A-Middle) Efficient Transformer architectures that came out with the LRA benchmark. (A-Bottom) Notable models that came out after the release of the LRA benchmark. (B) Architectures using Structured State Spaces. here the symbol † indicates that the results came from the original paper from the LRA dataset (Tay et al., 2020c). Furthermore, the symbol × indicates that the Transformer variant either ran out of memory and − indicates that results were not reported.

| Model | Year | ListOps | Text | Retrieval | Image | Pathfinder | Path-X | Avg. |
|---|---|---|---|---|---|---|---|---|
| Random | | 10.00 | 50.00 | 50.00 | 10.00 | 50.00 | 50.00 | 36.67 |
| | | | | (A) Transformer Based Architectures | | | | |
| Transformer† (Vaswani et al., 2017) | 2017 | 36.37 | 64.27 | 57.46 | 42.44 | 71.40 | × | 53,66 |
| Transformer (re-impl (Ma et al., 2021)) | 2017 | 37.11 | 65.21 | 79.14 | 42.94 | 71.83 | × | 57.71 |
| Local Att.† (Tay et al., 2020c) | 2017 | 15.82 | 52.98 | 53.39 | 41.46 | 66.63 | × | 46.71 |
| Sparse Trans.† (Child et al., 2019) | 2019 | 17.07 | 63.58 | 59.59 | 44.24 | 71.71 | × | 51.03 |
| Performer† (Choromanski et al., 2020) | 2020 | 18.01 | 65.40 | 53.82 | 42.77 | 77.05 | × | 51.18 |
| Reformer† (Kitaev et al., 2020) | 2020 | 37.27 | 56.10 | 53.40 | 38.07 | 68.50 | × | 50.56 |
| Sinkhorn Trans.† (Tay et al., 2020b) | 2020 | 33.67 | 61.20 | 53.83 | 41.23 | 67.45 | × | 51.23 |
| BigBird† (Zaheer et al., 2021) | 2021 | 36.05 | 64.02 | 59.29 | 40.83 | 74.87 | × | 54.18 |
| FNet (Lee-Thorp et al., 2021) | 2021 | 35.33 | 65.11 | 59.61 | 38.67 | 77.80 | × | 54,42 |
| Luna-16 (Ma et al., 2021) | 2021 | 37.43 | **65.74** | **79.38** | 46.39 | **78.36** | - | **59.55** |
| CAST Top-K (**Ours**) | 2023 | 39.90 | 65.45 | 78.01 | 52.37 | 70.18 | × | 59.32 |
| CAST SA Top-K (**Ours**) | 2023 | **40.70** | 65.13 | 74.64 | **52.78** | 62.22 | × | 57.57 |
| | | | | (B) Structured State Space Architectures | | | | |
| S4 (Gu et al., 2022) | 2021 | 59.60 | 86.82 | 90.90 | 88.65 | 94.20 | 96.35 | 86.09 |
| S5 (Smith et al., 2023) | 2022 | 62.15 | 89.31 | 91.40 | 88.00 | 95.33 | **98.58** | 87.46 |
| MEGA-Chunk (Ma et al., 2023) | 2022 | 58.76 | 90.19 | 90.97 | 85.80 | 94.41 | 93.81 | 85.66 |
| MEGA (Ma et al., 2023) | 2022 | **63.14** | **90.43** | **91.25** | **90.44** | **96.01** | 97.98 | **88.21** |

## 5.3 CLUSTERING ABLATION

**Clustering mechanisms.** Figure 3 shows the difference in performance, memory footprint, and time efficiency between the clustering mechanisms *Top-K* and SA *Top-K* on the *Text* and *Image* tasks of LRA. In Figure 3d, it can be seen that the choice in the clustering mechanism slightly affects the resulting performance on the *Image* task at a cluster size of 128 and 256. It can be observed from Figure 3b and Figure 3e, that the cluster mechanism does not affect the Peak Memory Usage. Furthermore, it can be seen from Figure 3c and Figure 3f that the *Top-K* clustering mechanism is overall significantly faster than the SA *Top-K* clustering mechanism. The SA Top-K clustering mechanism in particular is much slower when using small cluster sizes on large input sequences, like for the *Text* task.

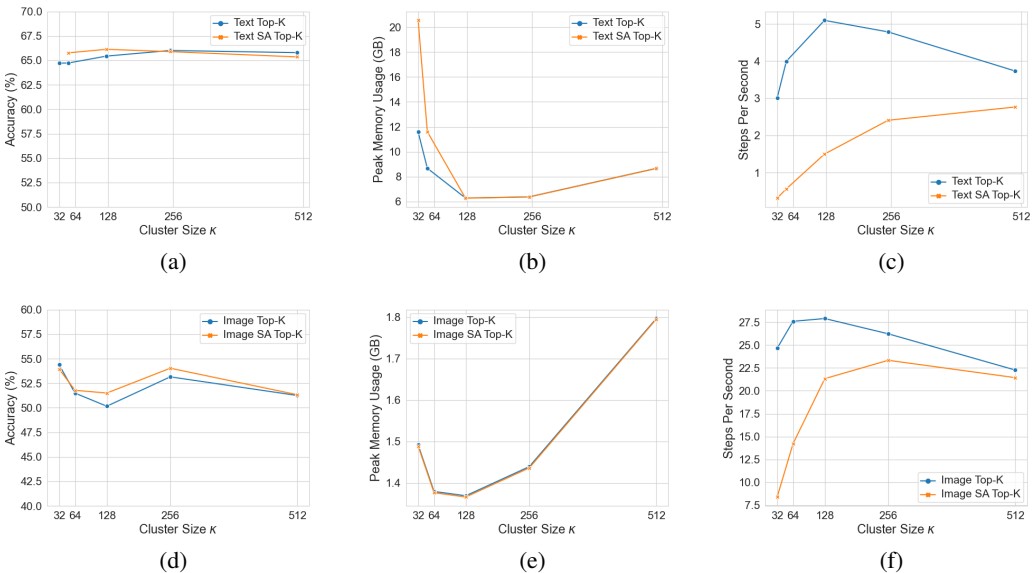

Figure 3: Ablations on the cluster size using CAST with *Top-K* Clustering Mechanism (blue) and Single Assignment *Top-K* Clustering Mechanism (orange) on the *Text* and *Image* tasks of the LRA benchmark against (a & d) the performance, (b & e) the peak memory allocated, and (c & f) the time efficiency, respectively.

**Performance.** In Figure 3a and Figure 3d, we show a comparison of the effect of cluster sizes and cluster mechanism on the performance of CAST on *Text* and *Image* task of the LRA dataset. For the *Text* task, the cluster size does not significantly impact the resulting accuracy, although a slight increase in accuracy can be observed at a larger cluster size. However, cluster size does impact the performance of the *Image* task significantly for both *Top-K* and SA *Top-K*. It can be observed that the performance on the *Image* task dips around a cluster size of 64 to 128, but peaks at a cluster size of 32 and 256.

**Peak memory usage.** In Figure 3b and Figure 3e, we show measurements of the influence of the cluster sizes on the peak memory usage of the *Image* and *Text* task. At its lowest, CAST only uses around 1.35 gigabytes of memory for the *Image* and 6.3 gigabytes of memory for the *Text* task. The memory curves represent a quadratic relationship, with an increase in memory when the number of clusters becomes too large, which was expected from Section 3.4. For both tasks, it can be seen that the least amount of memory is used when the number of clusters and the cluster size is close to the relation $N_c^2 = \kappa$. However, knowing that CAST achieves similar performance across different cluster sizes, we can use the cluster size that minimizes the memory footprint without a large decrease in performance.

**Time efficiency.** In Figure 3c and Figure 3f, we report measurements of the influence of the cluster size and clustering mechanism on the training steps per second of the *Text* and *Image* task. It can be observed that the number of training steps per second for the SA *Top-K* clustering mechanism (orange) is significantly lower than that of the standard *Top-K* clustering mechanism, especially at smaller cluster sizes. This is due to the constraint of SA *Top-K* must ensure every token is contained only in one cluster. However, knowing that the change of performance between clustering mechanisms and cluster sizes is small, the *Top-K* clustering mechanism can be chosen at a cluster size that maximizes the number of steps per second.

## 5.4 VISUAL ANALYSIS ON CLUSTERS

The clusters created by CAST do seem to hold visuospatial information on image tasks. More specifically, CAST seems to separate background from foreground in images. In Figure 6a, we show an example of an input image from the *Image* task which depicts a horse and its rider. In Figure 4b, the clustered pixels of the two layers of CAST are depicted, where each color corresponds to one of the clusters. In the first layer, we can observe that the clusters are approximately slices of the

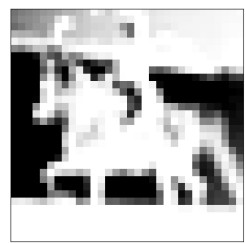 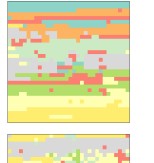 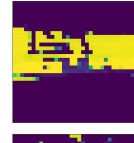 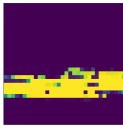 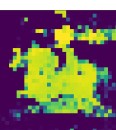 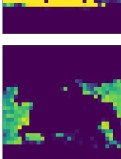

(a) Image of the LRA **Image** task.          (b) Visualizations of learned clusters of CAST per layer.

Figure 4: Visualizations of the learned clusters of a CAST model with SA *Top-K* on the LRA *Image* task. The number of clusters is 8. (a) An example image. (b- Left) Clustered pixels, where each color represents a different cluster. Example scores for clusters of $\mathbf{A}_g$ (b- Middle & Right), each image corresponds to a different cluster for the first (b-Top) and last layer (b-Bottom), respectively.

original image. In the last layer, it can be observed that the background and foreground of the image are roughly separated in different clusters. This behavior is observed for most of the images in the *Image* task – see Appendix A.6.3 for more examples. We further analyze the clusters by visualizing the scores of $\mathbf{A}_g$ in Figure 4b, where the separation of foreground and background is more evident, together with the separation per slice of the image.

## 5.5 LIMITATIONS

CAST is significantly faster than other methods based on efficient transformers. In terms of benchmark results, however, efficient transformers including CAST perform lower than Structured State Space models. While a direct comparison is unfair, as the architecture and working principle of these methods are different from transformers, with higher complexity, we report their results for the sake of completeness. A current limitation of CAST is the absence of a decoding version for generative natural language. While the focus of the paper is however on the optimization of the attention computation via a novel clustering of surrogate tokens approaches, we foresee that CAST could be adapted using asymmetric clustering and casual masking to create a decoder and be deployed in generative models as well.

## 6 CONCLUSIONS

We present CAST, a more efficient drop-in replacement for self-attention, which lowers the complexity of computing the self-attention in Transformers. The solution that we propose is based on clustering surrogate tokens, a novel approach in which the cluster directions are learnable, in contrast with static, algorithmically defined cluster directions of previous works. While our contribution is potentially general, and applicable to many tasks, we focus on analyzing its impact towards improving efficient computation of self-attention in transformers, especially at inference time, while maintaining the high results of standard transformer architectures. Our experiments demonstrate that the memory and computational efficiency of CAST is significantly better than other efficient Transformers (e.g. Reformer (Kitaev et al., 2020), Performer (Choromanski et al., 2020)) in the Long Range Arena efficiency benchmark. CAST uses less memory than existing methods for all tested sequence lengths, being up to about 6×faster than and using 10% of the memory of the original Transformer. Our future work will explore increasing the efficiency of CAST by parallelizing attention within clusters, and finding more efficient clustering mechanisms.

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

## A  APPENDIX

### A.1  MODULARIZED VISUALISATION OF CAST

In Figure 5, a modularized verison of the visualization of CAST can be seen. In Figure 5a, we apply Intra Cluster Attention by clustering the input $X$ based on the values in $A_g$, and then apply self-attention within the created clusters. In Figure 5b, we create the cluster summaries $R_{inter}$, which is based on a sum values weighted based on the the key-surrogate token similarity matrix $A_k$. In Figure 5c, we lastly combine the results $R_{intra}$ and $R_{inter}$ based on the query-surrogate token similarity matrix $A_q$, resulting in the output of CAST.

### A.2  NOMENCLATURE

We include a nomenclature in Table 3 to aid in the reading of Section 3.2. Here the symbols are given together with what they are representing.

Table 3: A nomenclature of symbols used in the equations defining CAST.

| Symbol | Meaning/Representation |
|---|---|
| $A_q$ | The dot product similarity between the queries ($Q$) and the surrogate tokens ($S$). |
| $A_k$ | The dot product similarity between the keys(K) and the surrogate tokens ($S$). |
| $\phi$ | A learned value similar to the queries/keys, represents whether it is more important to share or receive information. |
| $A_g$ | The combined similarity of $A_q$ and $A_k$, where $A_q$ has more weight if phi is high, and $A_k$ has more weight when phi is low. Used as the basis for clustering. |
| $X_g, Q_g, K_g, V_g$ | The clustered tokens, queries, keys, and values. |
| $R_intra$ | The result of self-attention within each cluster. |
| $A_value$ | The weights for the weighted sum of the cluster summaries. |
| $R_inter$ | The cluster summaries. |

### A.3  DETAILS OF THE CLUSTERING MECHANISMS

In this section, we describe our proposed *Top-K* clustering mechanism and the SA *Top-K* clustering mechanism.

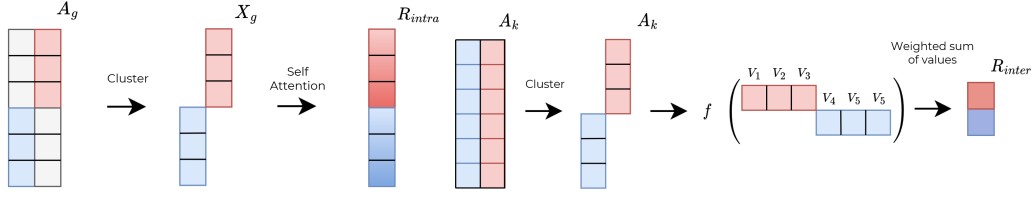

(a) Intra Cluster Attention

(b) Creation of the cluster summaries $R_{inter}$, which is a weighted sum of each cluster's values based on the weights in $A_k$.

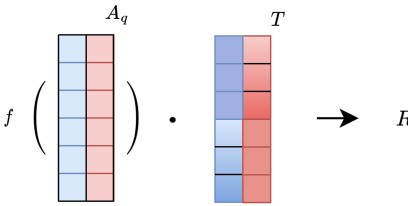

(c) The combining of $R_{inter}$ and $R_{intra}$ using $A_q$ as the weights for the weighted sum.

Figure 5: A modularized sketch of the proposed method. Here, some details are omitted to make it easier to read. (a) shows intra-cluster self-attention, (b) shows the creation of the cluster summaries $R_{inter}$, and (c) shows how $R_{inter}$ and $R_{intra}$ are combined.

### A.3.1 TOP-K CLUSTERING

The *Top-K* clustering mechanism groups the indices with the largest similarity scores in $\mathbf{A}_g$, it allows for a token to be clustered into two clusters, but also for a token not to be clustered at all. A formal definition of the *Top-K* clustering mechanism is in Algorithm 1, where $A \in \mathbb{R}^{N \times N_c}$ is the similarity scores for each token to each cluster, and $X \in \mathbb{R}^{N \times *}$ is a matrix of feature vectors that we wish to cluster, where $*$ indicates any shape.

---

**Algorithm 1** Implementation of the proposed *Top-K* clustering mechanism.

**Input** X, A
**Output** C
1: **function** SA *Top-K*$(A, X)$
2:      $C = \{C_1...C_{N_c}\}$                        ▷ Initialize result
3:      $I, A_{top} = \text{Top-K}(A)$         ▷ Get the indices of the largest values per cluster
4:      **for** $i \leftarrow 1$ to $N_c$ **do**
5:          **for** $j \leftarrow 1$ to $\frac{N}{N_c}$ **do**
6:              $i_{token} = I_{i,j}$
7:              $C_i.\text{insert}(X_{i_{token}})$
8:          **end for**
9:      **end for**
10: **end function**

---

### A.3.2 SINGLE ASSIGNMENT TOP-K CLUSTERING

The single assignment *Top-K* clustering mechanism has the constraint that each token is assigned to only a single cluster, while also maximizing the total similarity for all clusters combined. The SA *Top-K* clustering mechanism is formally defined in Algorithm 2, where $A \in \mathbb{R}^{N \times N_c}$ represents the similarity scores for each token to each cluster, and $X \in \mathbb{R}^{N \times *}$ represents a matrix of feature vectors that we wish to cluster.

---

**Algorithm 2** Implementation of the proposed Single Assignment *Top-K* clustering mechanism.

---

    **Input** X, A
    **Output** C
1: **function** SA *Top-K*$(A, X)$
2:     $A^c, I^c = sort_2(A)$                             ▷ Sort from highest to lowest cluster
3:     $A^r, I^r = sort_1(A^c)$                         ▷ Sort from highest to lowest token
4:     $C = \{C_1 ... C_{N_c}\}$                               ▷ Initialize result
5:     $M = \mathbf{0}^N$                                 ▷ Initialize Assignment Mask
6:     **for** $i \leftarrow 1$ to $N_c$ **do**
7:         **for** $j \leftarrow 1$ to $N$ **do**
8:             $j_{token} = I^r_j$
9:             $i_{cluster} = I^c_{j_{token}}$
10:             **if** $M_j = 1$ **or** $length(C_{i_{cluster}}) = \frac{N}{N_c}$ **then**
11:                 continue for loop
12:             **end if**
13:             $C_{i_{cluster}}.insert(X_{j_{token}})$
14:             $M_{j_{token}} = 1$
15:         **end for**
16:     **end for**
17:     **return** C
18: **end function**

---

### A.4   Long Range Arena Benchmark

The Long Range Arena (LRA) benchmark contains six tasks [ListOps, Text, Retrieval, Image, Path, and Path-X] that represent a diverse and intricate spectrum of challenges. Each task demands a distinct set of skills, ranging from semantic understanding and reasoning to image comprehension and logical manipulation. This diverse selection of tasks aims to assess the capabilities of architectures, ensuring a comprehensive evaluation that goes beyond singular skill acquisition. Furthermore, the LRA benchmark holds a unique position within the research community as it is the common benchmark for comparing efficiency and performance, such as for the architectures in Luna (Ma et al., 2021), MEGA (Ma et al., 2023), and S4 (Gu et al., 2022). This widespread adoption signifies a consensus among researchers regarding its suitability for assessing the performance of experimental frameworks, including our proposed CAST. Next, we describe how these six tasks are treated.

**ListOps.** The ListOps dataset was created for testing the parsing ability of latent tree models, but a larger version is now used in the LRA to test the capability of Transformers to learn the hierarchical structures. The data is a sequence of tokens representing a large mathematical operation on lists of numbers. The numbers 0 to 9 are available as both the input of the operations and the final result. There are four base mathematical operations :

- MAX: The largest value in a given list.

- MIN: The smallest value in a given list.

- MED: The median value in a given list.

- SUM MOD: The sum of the list module 10.

In the LRA the maximal length of the input sequence is set to 2K tokens. This is a ten-way classification task where accuracy is used as the evaluation metric.

**Text.** The **Text** task takes the IMDb reviews sentiment classification task (Maas et al., 2011) and the characters as tokens in the input sequence. The maximum length of the input sequences is truncated or padded to 4K tokens. This task is a binary classification task with accuracy as its metric.

**Retrieval.** For the **Retrieval** task the ACL Anthology Network dataset (Radev et al., 2013) is used. For this dataset, the task is to determine whether two papers are linked by a citation. Both papers are passed to the Transformer variant, creating compressed representations, which are then combined and passed into a classification head. With this setup the **Retrieval** task can be considered a binary classification task. To make the task more challenging, character-level tokens like in the text

classification task are used in the setup. A sequence length of 4K tokens is used per document the use of 8K tokens per example.

**Image.** The **Image** task takes the CIFAR-10 dataset (Krizhevsky, 2009) as its base. The images are first greyscaled into a single channel with each pixel having an 8-bit pixel intensity as its representation. This results in a $32 \times 32$ image which is unrolled into a 1-D sequence, this sequence is then used as input for a ten-way classification task.

**Pathfinder.** The **Pathfinder** task (Linsley et al., 2018) consists of images of $32 \times 32$ where two dots, represented by circles, are connected by dashed lines. A model is required to make a binary decision of whether the two dots are connected by the dashed lines, however, there are also distraction lines that are not connected to any of the dots. Just like in the image classification task the image is unrolled into a sequence length of 1024 and used as input for this task.

**Path-X.** The **Path-X** task is a more extreme case of the original Pathfinder, instead of the image being $32 \times 32$ it is $128 \times 128$ making the sequence length 16 times larger than the original. Apart from the size this task is exactly the same as the original Pathfinder. It should be noted that this task has not yet been achieved with a higher-than-random accuracy with the constraints of the LRA.

## A.5 EXPERIMENT DETAILS

For all tasks, we follow the standards given in the original Long Range Arena paper (Tay et al., 2020c) regarding the data processing and task setup. For our choices in most hyperparameters, we used the current state-of-the-art, MEGA (Ma et al., 2023), as our baseline regarding the number of weight updates. Furthermore, we use their data splits regarding all tasks. The final hyperparameters used for our reported accuracy are in Table 4. For both the reported performance of *Top-K* and SA *Top-K* the same hyperparameters are used. General hyperparameters include the averaging of the output features over the sequence for the classification features, the use of linear feature embeddings for pixel tasks, the use of sinusoidal positional embeddings for all tasks, and an extra normalization layer on the output features when pre-normalization is used.

Table 4: Final hyperparameters for the best performing CAST models in our hyperparameter sweep. Here, Depth indicates the number of Transformer blocks, $h$ the number of heads, $d$ the number of features in the self-attention block, $d_{ff}$, the number of features in the feedforward block, $d_{emb}$ the number of features in the embedding, $N_c$ the number of clusters, Norm the type of normalization being used, BS the batch-size, LR the learning rate, WD the weight decay, and Epochs the number of epochs that were trained for.

| Task | Depth | $h$ | $d$ | $d_{ff}$ | $d_{emb}$ | $N_c$ | Norm | Pre-norm | BS | LR | WD | Epochs |
|------|-------|-----|-----|----------|-----------|-------|------|----------|-----|-----|-----|--------|
| ListOps | 4 | 8 | 64 | 128 | 256 | 10 | Layer | False | 64 | $1e^{-3}$ | $1e^{-2}$ | 60 |
| Text | 4 | 4 | 64 | 128 | 256 | 20 | Scale | False | 25 | $1e^{-3}$ | $1e^{-2}$ | 25 |
| Retrieval | 2 | 8 | 256 | 256 | 256 | 20 | Layer | False | 8 | $1e^{-2}$ | $1e^{-2}$ | 5 |
| Image | 2 | 2 | 128 | 128 | 256 | 16 | Batch | True | 50 | $5e^{-3}$ | $1e^{-2}$ | 200 |
| Pathfinder | 2 | 2 | 32 | 32 | 64 | 16 | Batch | True | 128 | $1e^{-3}$ | $1e^{-2}$ | 200 |

## A.6 DETAILED RESULTS

In this section, we go into more depth regarding the results of CAST. We first give an extensive overview of other long-range sequence modeling architectures, and then show more examples of the visual analysis that was done on the **Image** task.

### A.6.1 LONG RANGE ARENA EFFICIENCY

We further compare the relative efficiency of CAST Top-K with the vanilla Transformer during inference time in Table 5. Here, we observe that during inference CAST is still significantly faster and more memory efficienct than the vanilla Transformer.

### A.6.2 LONG RANGE ARENA PERFORMANCE

In Table 6, we report a detailed list of results of different efficient Transformer variants, and other long-range sequence models on the Long Range Arena benchmark. We divide these models into the following categories: Transformer Based Architectures, Structured State Space Architectures, and Other Architectures. We can see that among the Transformer Based Architectures (A) Luna

Table 5: Speed and Memory efficiency of the LRA Benchmark during inference.

| Model | Steps Per Second ↑ | | | | Peak Memory Usage ↓ | | | |
|---|---|---|---|---|---|---|---|---|
| | 1K | 2K | 3K | 4K | 1K | 2K | 3K | 4K |
| Transformer | 1.0 | 1.0 | 1.0 | 1.0 | 1.0 | 1.0 | 1.0 | 1.0 |
| CAST (Top-K) | 1.87 | 3.95 | 5.27 | 6.91 | 0.280 | 0.150 | 0.102 | 0.081 |

(Ma et al., 2021) and CAST similarly strong performance. When it comes to the Structured State Space Architectures (B), it can be observed that all models perform similarly, with MEGA (Ma et al., 2023) being slightly better than the rest. As for the other types of architectures, they neither use self-attention nor structured state spaces to "mix" their input sequence. Among them, the recent ChordMixer (Khalitov et al., 2023) stands out, ChordMixer was created for handling data with extremely long sequence lengths (in the order of 100K tokens), but has shown impressive results on the LRA benchmark too.

Table 6: The performance of different architectures on the Long Range Arena benchmark in classification accuracy. We divide these works into (A) Transformer architectures that do not use Structured State Spaces or any derivation of this, (B) Architectures using Structured State Spaces, and (C) Other types of architectures. The (A)-related models are grouped as; (A-Top) The original Transformer architecture, (A-Middle) efficient Transformer architectures that came out with the LRA benchmark, and (A-Bottom) notable models that came out after the release of the LRA benchmark. Here the symbol † indicates that the results came from the original paper from the LRA dataset (Tay et al., 2020c). Furthermore, the symbol × indicates that the Transformer variant either ran out of memory and − indicates that results were not reported.

| Model | Year | ListOps | Text | Retrieval | Image | Pathfinder | Path-X | Avg. |
|---|---|---|---|---|---|---|---|---|
| Random | | 10.00 | 50.00 | 50.00 | 10.00 | 50.00 | 50.00 | 36.67 |
| | | (A) Transformer Based Architectures | | | | | | |
| Transformer[†] (Vaswani et al., 2017) | 2017 | 36.37 | 64.27 | 57.46 | 42.44 | 71.40 | × | 53.66 |
| Transformer (re-impl (Ma et al., 2021)) | 2017 | 37.11 | 65.21 | 79.14 | 42.94 | 71.83 | × | 57.71 |
| Sparse Trans.[†] (Child et al., 2019) | 2019 | 17.07 | 63.58 | 59.59 | 44.24 | 71.71 | × | 51.03 |
| Local Att.[†] (Tay et al., 2020c) | 2020 | 15.82 | 52.98 | 53.39 | 41.46 | 66.63 | × | 46,71 |
| Reformer[†] (Kitaev et al., 2020) | 2020 | 37.27 | 56.10 | 53.40 | 38.07 | 68.50 | × | 50.56 |
| Sinkhorn Trans.[†] (Tay et al., 2020b) | 2020 | 33.67 | 61.20 | 53.83 | 41.23 | 67.45 | × | 51.23 |
| Performer[†] (Choromanski et al., 2020) | 2020 | 18.01 | 65.40 | 53.82 | 42.77 | 77.05 | × | 51.18 |
| Linformer[†] (Wang et al., 2020) | 2020 | 35.70 | 53.94 | 52.27 | 38.56 | 76.34 | × | 51.36 |
| Longformer[†] (Beltagy et al., 2020) | 2020 | 35.63 | 62.85 | 56.89 | 42.22 | 69.71 | × | 53.46 |
| Synthesizer[†] (Tay et al., 2020a) | 2021 | 36.99 | 61.68 | 54.67 | 41.61 | 69.45 | × | 52.40 |
| BigBird[†] (Zaheer et al., 2021) | 2021 | 36.05 | 64.02 | 59.29 | 40.83 | 74.87 | × | 54.18 |
| Luna-16 (Ma et al., 2021) | 2021 | 37.43 | 65.74 | **79.38** | 46.39 | 78.36 | - | **59.55** |
| Luna-128 (Ma et al., 2021) | 2021 | 38.01 | 65.74 | 79.55 | 47.47 | **78.89** | - | 59,94 |
| Luna-256 (Ma et al., 2021) | 2021 | 37.98 | **65.78** | 79.56 | 47.86 | 78.55 | - | 59,96 |
| PSF (Khalitov et al., 2021) | 2021 | 38.85 | 77.32 | - | 45.01 | 80.49 | - | 56.95 |
| FNet (Lee-Thorp et al., 2021) | 2021 | 35.33 | 65.11 | 59.61 | 38.67 | 77.80 | × | 54.42 |
| CAST Top-K (**Ours**) | 2023 | 39.90 | 65.45 | 78.01 | 52.37 | 70.18 | × | 59.32 |
| CAST SA Top-K (**Ours**) | 2023 | **40.70** | 65.13 | 74.64 | **52.78** | 62.22 | × | 57.57 |
| | | (B) Structured State Space Architectures | | | | | | |
| S4 (Gu et al., 2022) | 2021 | 59.60 | 86.82 | 90.90 | 88.65 | 94.20 | 96.35 | 86.09 |
| H3 (Dao et al., 2023) | 2022 | 57.50 | 88.20 | 91.00 | 87.30 | 93.00 | 91.80 | 84.80 |
| Liquid-S4 (Hasani et al., 2022) | 2022 | 62.75 | 89.02 | 91.20 | 89.50 | 94.8 | 96.66 | 87.32 |
| SGConv (Li et al., 2022) | 2022 | 61.45 | 89.20 | 91.11 | 87.97 | 95.46 | 97.83 | 87.17 |
| S5 (Smith et al., 2023) | 2022 | 62.15 | 89.31 | 91.40 | 88.00 | 95.33 | **98.58** | 87.46 |
| MEGA-Chunk (Ma et al., 2023) | 2022 | 58.76 | 90.19 | 90.97 | 85.80 | 94.41 | 93.81 | 85.66 |
| MEGA (Ma et al., 2023) | 2022 | **63.14** | **90.43** | **91.25** | **90.44** | **96.01** | 97.98 | **88.21** |
| | | (C) Other Architectures | | | | | | |
| Paramixer (Yu et al., 2022) | 2022 | 39.57 | 83.32 | - | 46.58 | 80.49 | - | 58.33 |
| CDIL (Cheng et al., 2022) | 2022 | - | 87.61 | 84.27 | 64.49 | 91.00 | - | 64.56 |
| ChordMixer (Khalitov et al., 2023) | 2023 | 60.12 | **88.82** | **89.98** | **90.17** | **96.69** | **98.63** | **87.40** |

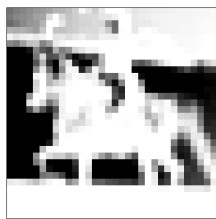

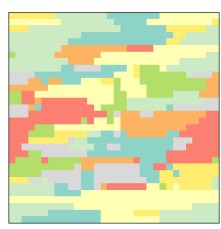

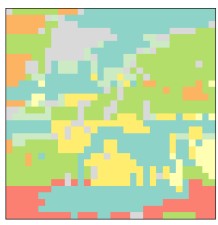

(a) Example Image of the LRA **Image** task.

(b) Clustered Image, 8 hashes, first layer.

(c) Clustered Image, 8 hashes, last layer.

Figure 6: A visualization of learned clusters when using LSHAttention. Here 6a shows the original input image, 6b shows the way pixels are clustered in the first layer of LSHAttention, and 6c shows the way pixels are clustered in the last layer of LSHAttention

### A.6.3 FURTHER VISUAL ANALYSIS

Additional visualizations of the clusters created for different samples from the **Image** task of LRA, can be seen in Figure 7 (a horse), Figure 8 (a deer), and Figure 9 (an automobile). For each of these figures, subfigure (a) shows the original input image, (b) shows the assignment of clusters for each pixel in the first layer, and (c) shows the assignment of clusters for each pixel in the last layer. Subfigure (d) shows for each cluster the score in $\mathbf{A}_g$ that each token had for the first layer. Subfigure (e) shows for each cluster the score in $\mathbf{A}_g$ that each pixel had for the last layer.

For the mentioned sample images, it can be seen that the in the first layer, i.e. in Figures 7d, 8d, and Figure 9d, each cluster roughly clusters the same pixels. This behavior could occur, because the positional embeddings are most prominent in the first layer, causing the surrogate tokens to cluster based on this positional embedding. Furthermore, it also shows that CAST learns to cluster slices of the image first, similar to convolution. In Figures 7e, 8e and, Figure 9e, the scores in $\mathbf{A}_g$ for the last layer can be seen. This layer (e) shows more image-specific clustering. For instance, from these scores, we can observe the outline and inverse outline of a horse, a deer in a forest, and an automobile, respectively. We interpret this as the separation of background and foreground. In the case of the deer, Figure 8e, we observe a more rough outline, which can be due to the fact that the background and foreground of this image are much more similar.

### A.6.4 REFORMER VISUAL ANALYSIS

To determine whether the visuospatial information contained in the clustering of CAST is general feature of clustering Transformers like the Reformer and SMYRF, we trained a Reformer model on the CIFAR-10 dataset and inspected the clustered images. In Figure 6, an example image for clustering created by the Reformer's LSHAttention can be seen.

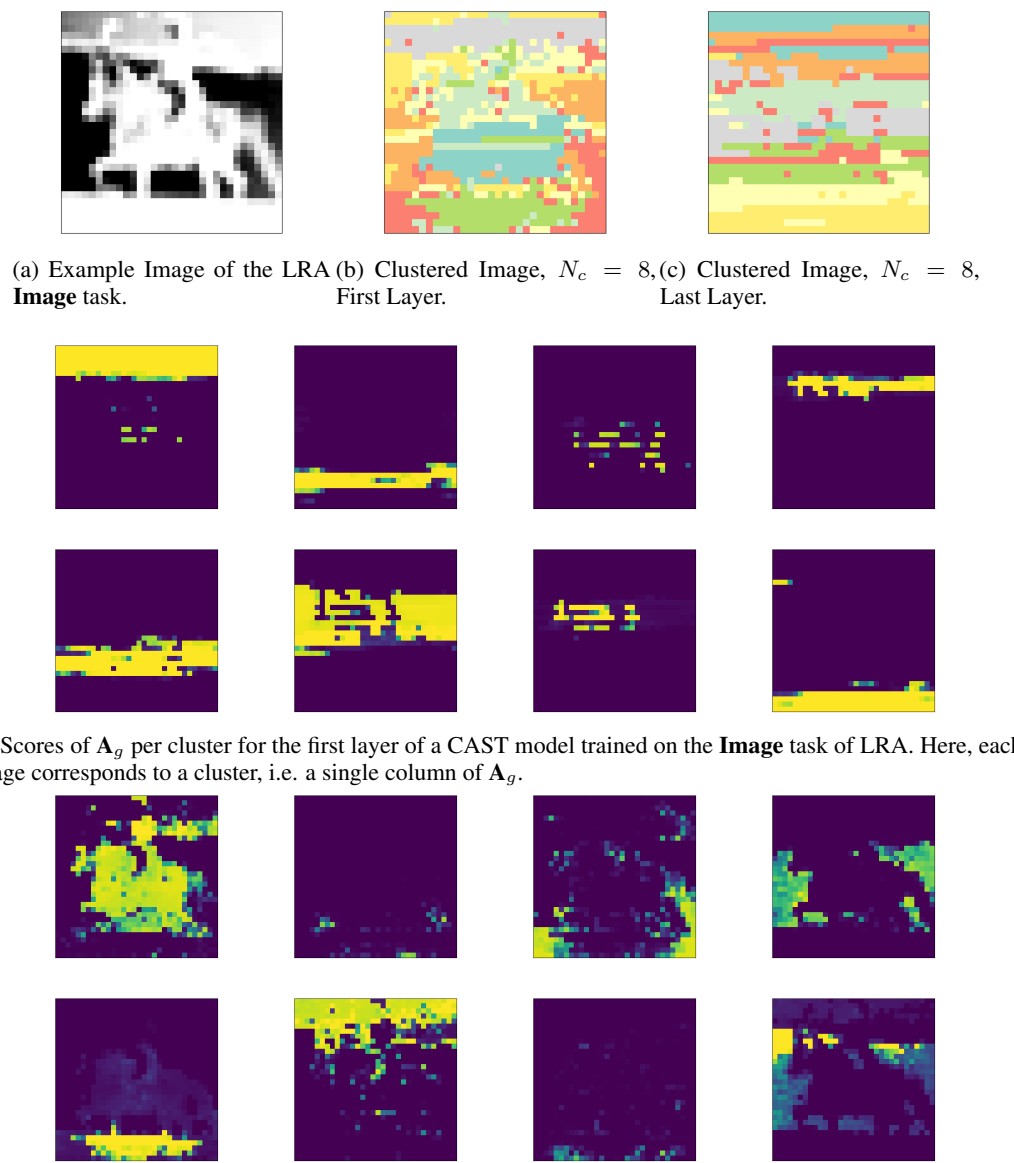

(a) Example Image of the LRA
**Image** task.

(b) Clustered Image, $N_c = 8$,
First Layer.

(c) Clustered Image, $N_c = 8$,
Last Layer.

(d) Scores of $\mathbf{A}_g$ per cluster for the first layer of a CAST model trained on the **Image** task of LRA. Here, each image corresponds to a cluster, i.e. a single column of $\mathbf{A}_g$.

(e) Scores of $\mathbf{A}_g$ per cluster for the last layer of a CAST model trained on the **Image** task of LRA. Here, each image corresponds to a cluster, i.e. a single column of $\mathbf{A}_g$.

Figure 7: A visualization learned clusters in different layers of CAST. Here, (a) is a sample from the **Image** of LRA, depicting a horse with a rider, (b) is an image representing the clustered pixels in the first layer of CAST, (c) is an image representing the clustered pixels in the last layer of CAST, (d) the scores in $\mathbf{A}_g$ for every pixel in each of the eight clusters in the first layer of CAST, (e) the scores in $\mathbf{A}_g$ for every pixel in each of the eight clusters in the last layer of CAST.

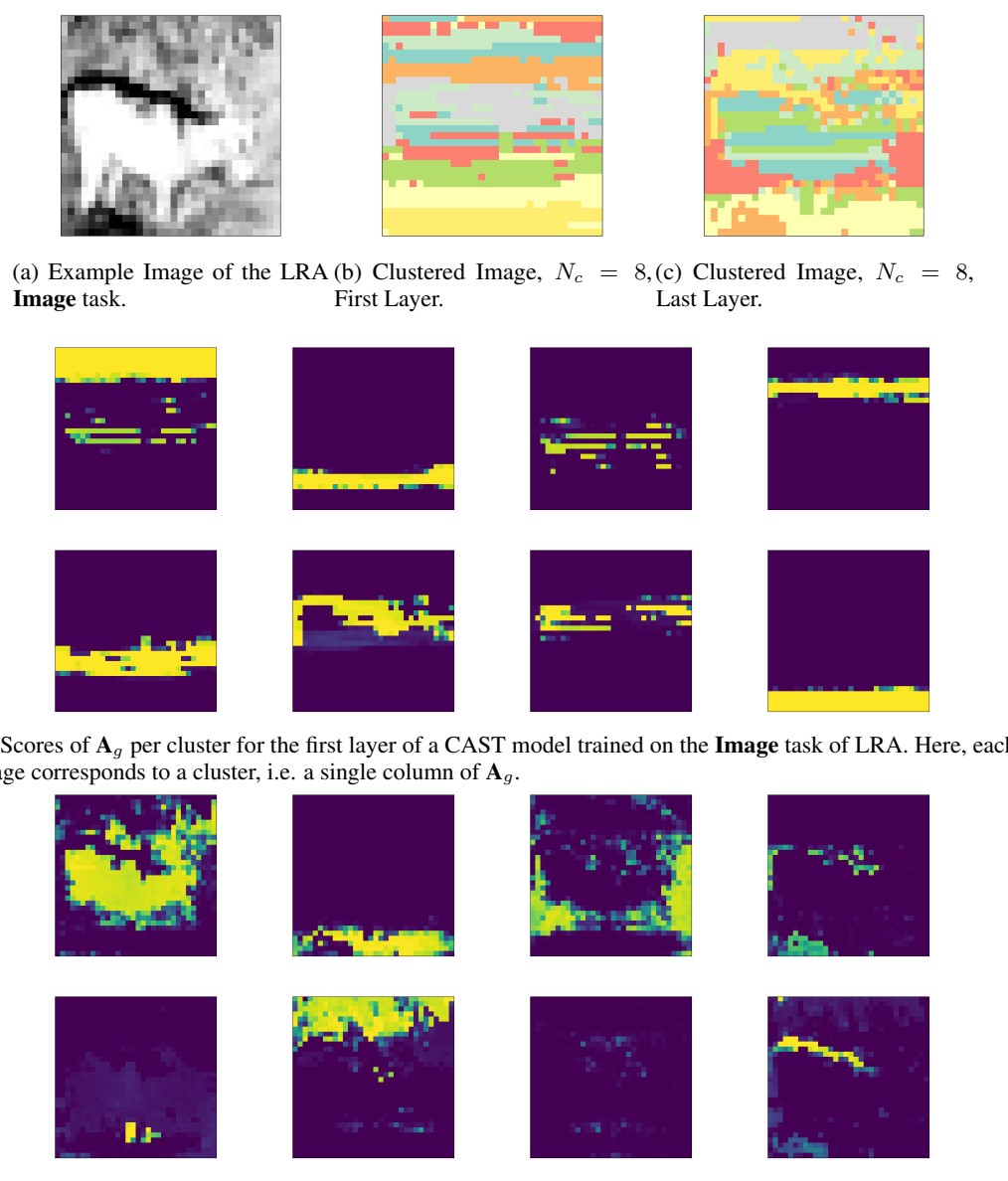

(a) Example Image of the LRA **Image** task.

(b) Clustered Image, $N_c = 8$, First Layer.

(c) Clustered Image, $N_c = 8$, Last Layer.

(d) Scores of $\mathbf{A}_g$ per cluster for the first layer of a CAST model trained on the **Image** task of LRA. Here, each image corresponds to a cluster, i.e. a single column of $\mathbf{A}_g$.

(e) Scores of $\mathbf{A}_g$ per cluster for the last layer of a CAST model trained on the **Image** task of LRA. Here, each image corresponds to a cluster, i.e. a single column of $\mathbf{A}_g$.

Figure 8: A visualization of the learned clusters in different layers of CAST. Here, (a) is a sample from the **Image** of LRA, depicting a deer in a forest, (b) is an image representing the clustered pixels in the first layer of CAST, (c) is an image representing the clustered pixels in the last layer of CAST, (d) the scores in $\mathbf{A}_g$ for every pixel in each of the eight clusters in the first layer of CAST, (e) the scores in $\mathbf{A}_g$ for every pixel in each of the eight clusters in the last layer of CAST.

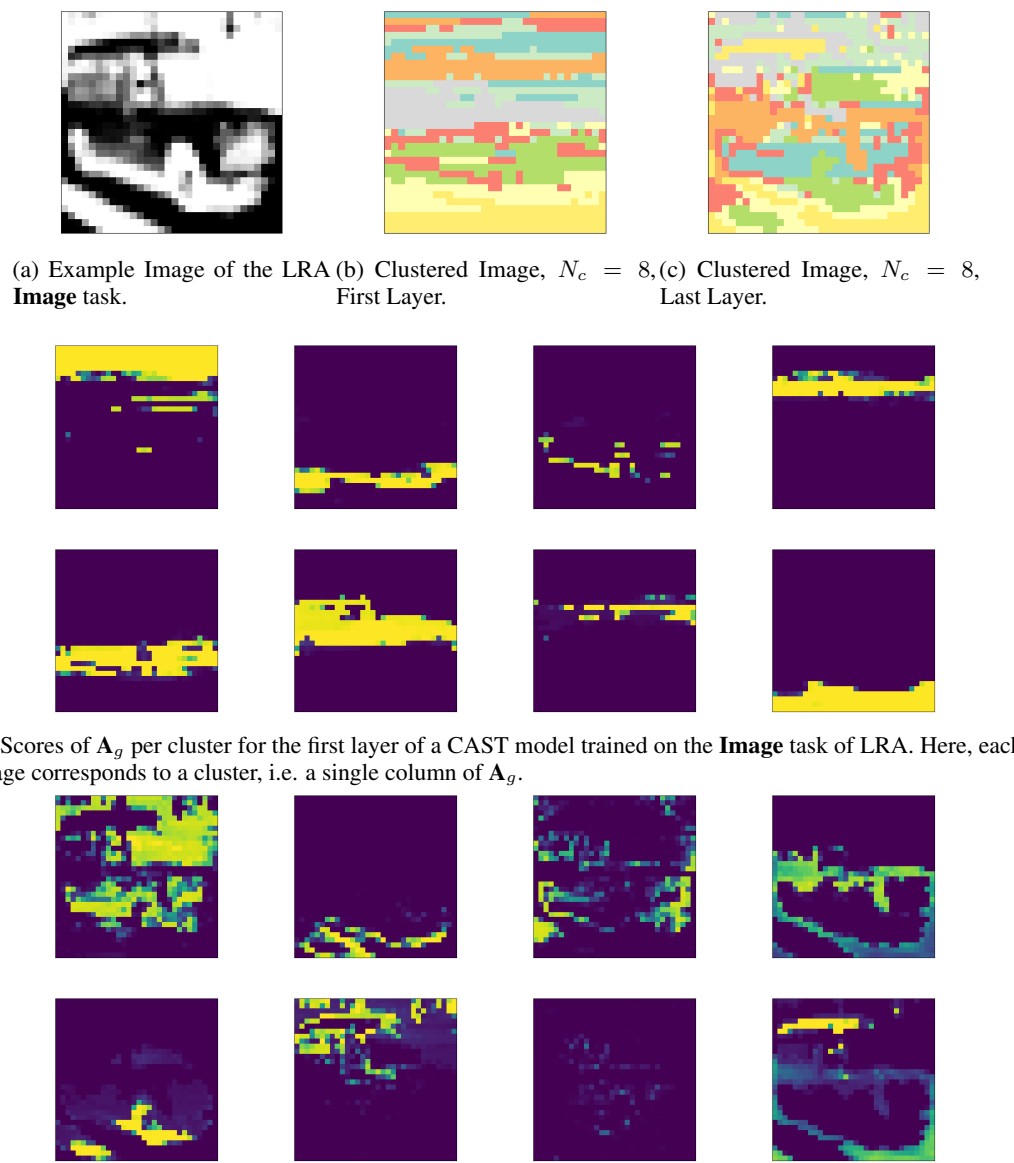

(a) Example Image of the LRA **Image** task.

(b) Clustered Image, $N_c = 8$, First Layer.

(c) Clustered Image, $N_c = 8$, Last Layer.

(d) Scores of $\mathbf{A}_g$ per cluster for the first layer of a CAST model trained on the **Image** task of LRA. Here, each image corresponds to a cluster, i.e. a single column of $\mathbf{A}_g$.

(e) Scores of $\mathbf{A}_g$ per cluster for the last layer of a CAST model trained on the **Image** task of LRA. Here, each image corresponds to a cluster, i.e. a single column of $\mathbf{A}_g$.

Figure 9: A visualization learned clusters in different layers of CAST. Here, (a) is a sample from the **Image** of LRA, depicting an automobile, (b) is an image representing the clustered pixels in the first layer of CAST, (c) is an image representing the clustered pixels in the last layer of CAST, (d) the scores in $\mathbf{A}_g$ for every pixel in each of the eight clusters in the first layer of CAST, (e) the scores in $\mathbf{A}_g$ for every pixel in each of the eight clusters in the last layer of CAST.

