# OpenReview forum: "CAST: Clustering self-Attention using Surrogate Tokens for efficient transformers"
_ICLR.cc/2024/Conference — ICLR 2024 Conference Withdrawn Submission_

### Official Review · Reviewer_KR9t · 2023-10-30

**Soundness:** 2 fair
**Presentation:** 2 fair
**Contribution:** 2 fair
**Rating:** 3
**Confidence:** 5

**Summary:**

This paper proposes a new efficient self-attention mechanism to reduce the $O(N^2)$ cost to $O(\alpha N)$ where $\alpha$ depends on the number of clusters. The author introduce a set of learnable surrogate tokens and use them to cluster the input tokens into different clusters and use the surrogate tokens to generate cluster summaries. Then, the final attention output is a weighted sum of attention within each cluster (intra attention) and the summaries of the clusters (inter attention). The authors show the proposed method gives better performance on LRA benchmark compared to other efficient Transformers but is worse than structured state space models.

**Strengths:**

1. The inter attention and intra attention provides coarse global view and fine local view of the input sequence, respectively, while providing efficiency benefits.

2. The proposed method uses learnable surrogate tokens to determine the cluster assignment rather than relying on some fixed criteria (such as locality), making it more flexible.

3. The ablation study evaluates different choice of the clustering algorithms and different choice of cluster sizes to understand the behavior of the proposed algorithm.

**Weaknesses:**

1. Figure 1 does not convey the main idea clearly and should be significantly improved.

2. The presentation of the proposed method in 3.2 is confusing and should be significantly improved. For example, it would be better to have a small roadmap on the beginning of 3.2 so that the readers know what each step is doing. Also, it would be better to break up the page-long paragraph to smaller paragraphs, and use each paragraph to explain a small part of computation. Also, explain the intention of each equation and the reasons of design choice.

3. The related work only discussed sparse efficient attentions and Reformer and SMYRF that related to the proposed idea. There are a lot more efficient attentions that have the property of “information flow throughout the entire input sequence” (which was one of the motivation for the proposed idea), such as low rank based attentions (Linformer https://arxiv.org/abs/2006.04768, Nystromformer https://arxiv.org/abs/2102.03902, Performer https://arxiv.org/abs/2009.14794, RFA https://arxiv.org/abs/2103.02143) or multi-resolution based attentions (H-Transformer https://arxiv.org/abs/2107.11906, MRA Attention https://arxiv.org/abs/2207.10284).

4. Missing discussion about Set Transformer (https://arxiv.org/abs/1810.00825) and other related works that also uses summary tokens.

5. In 4-th paragraph of related work, the authors claim some baselines are unstable and the proposed method is stable, but the claim is not supported by any experiments.

6. Experiments are only performed on LRA benchmark, which consists a set of small datasets. The results might be difficult to generalize to larger scale experiments. It would be better to evaluate the methods on larger scale datasets, such as language modeling or at least ImageNet.

7. Given that LRA benchmark is a small scale experiment, it would be better to run the experiment multiple times and calculate the error bars since the results could be very noisy.

**Questions:**

1. $f(\cdot)$ is not defined clearly. In line 2 of page 4, $f(\cdot)$ is an attention function, but based on the context, $f(\cdot)$ seems to be the softmax function.

2. How does the efficiency of the proposed method compare to efficient implementation of full self-attention (FlashAttention), which was integrated in PyTorch 2.

---

### Official Review · Reviewer_L36F · 2023-10-31

**Soundness:** 2 fair
**Presentation:** 2 fair
**Contribution:** 2 fair
**Rating:** 5
**Confidence:** 4

**Summary:**

This paper proposes a linear self-attention, a clustering self-attention mechanism (CAST), by
learning surrogate tokens to reduce the quadratic complexity of standard self-attention.

The proposed CAST shows comparable performance to efficient Transformer variants on long-range modeling tasks with reduced
time and memory complexity.

**Strengths:**

The motivation to propose linear attention for reducing memory/computation cost is reasonable and the clustering Attention scheme is interesting.

Ablation study is conducted to support the efficiency of CAST.

**Weaknesses:**

Missing comparisons with some token-reduction based approximation methods, such as Nystromformer [1] and Clustered Attention [2].

Not enough experiments. LRA benchmark does not necessarily lead to high performance in real applications.
It will be interesting to see some comparisons on GLUE/SQuAD benchmarks for BERT approximation [1][2] and on ImageNet for ViT-Base
with different self-attention variants as in [3].

There is no theretical analysis to show how accurate of the CAST approxiamtion can be. What is the approximation error?
Without this, it is very hard to understand why this approximation can work.

The improvement over baseline is quite incremental. Even comparing with Luna-16 (2021) [4], CAST is not showing better performance on Luna-16.

CAST (Top-K) is better than CAST (SA Top-K) on LRA with improved efficiency. I do not understand why CAST (SA Top-K) is needed.


[1] Nyströmformer: A Nyström-Based Algorithm for Approximating Self-Attention
[2] Fast Transformers with Clustered Attention
[3] SOFT: Softmax-free Transformer with Linear Complexity
[4] Luna: Linear unified nested attention

**Questions:**

Please see the weaknesses. I will change my rating based on the rebuttal.

---

### Official Review · Reviewer_sYTK · 2023-11-05

**Soundness:** 1 poor
**Presentation:** 3 good
**Contribution:** 2 fair
**Rating:** 3
**Confidence:** 5

**Summary:**

This work proposes to reduce the computational requirements of self-attention through the use of learnable surrogate tokens to cluster the input sequence. The intra-cluster self-attention is combined with inter-cluster summaries to enable the flow of information between all tokens in a sequence. Two different cluster assignment schemes, namely Top-K and Single Assignment Top-K, are explored to partition tokens based on affinity scores $A_g$. Experiments are conducted on the Long Range Arena to understand the efficacy of the proposed method.

**Strengths:**

- While the idea of clustering to partition and reduce complexity has been explored before, the specific approach of mixing intra-cluster and inter-cluster information is novel.
- The writing is clear and generally easy to follow.

**Weaknesses:**

*Experimentation:*

- The work conducts experiments only on the Long Range Arena benchmark. While LRA is useful to understand inference/training times and performance to some extent, it also has limitations when it comes to understanding performance on real-world data [1,2]. Many efficient attention mechanisms perform nearly equally on LRA, thus making it a limited benchmark [2]. Besides experimentation with real data, more recent benchmarks such as CAB [1], along with LRA, would make for a more comprehensive test suite. At present, the experimentation is severely limited.
- The numbers reported for comparison against prior work such as Performer may not be directly comparable due to differences in GPUs.

*Prior work discussion:*

- It would be beneficial to include more discussions on closely related works that use clustering [3,4,5,6].



[1] CAB: Comprehensive Attention Benchmarking on Long Sequence Modeling, ICML 2023

[2] Simple Local Attentions Remain Competitive for Long-Context Tasks, ACL 2022

[3] Sparse Attention with Learning-to-Hash, ICLR 2021

[4] End-to-End Object Detection with Adaptive Clustering Transformer, BMVC 2021

[5] Fast Transformers with Clustered Attention, NeurIPS 2020

[6] Cluster-Former: Clustering-based Sparse Transformer for Long-Range Dependency Encoding, ACL 2021

**Questions:**

- What is the number of clusters in the Long Range Arena benchmarks? Are the chosen numbers dependent on the sequence length?
- How are the number of clusters and cluster size expected to scale with real data and across different tasks?
- The asymptotic complexity is stated as $O(\alpha N)$; however, for the case where $\kappa= \frac{N}{N_c}$, wouldn't the complexity be quadratic? Is it possible to achieve sub-quadratic complexity while ensuring each token is clustered?

---

### Official Review · Reviewer_y2Xn · 2023-11-06

**Soundness:** 3 good
**Presentation:** 3 good
**Contribution:** 2 fair
**Rating:** 5
**Confidence:** 4

**Summary:**

The paper presents a self-attention replacement named CAST. The proposed method extracts queries, keys and values as usual but instead of computing attention scores between the queries and keys it proposes to compute attention scores with a set of learnable tokens. Subsequently these attentions scores are used to separate the queries, keys and values to clusters where the standard attention computation is performed only inside each cluster. In addition, summaries of the values in each cluster are computed and combined with the intra cluster values using the keys to clustering tokens and queries to clustering tokens attentions respectively. Experiments on the Long Range Arena benchmark show that the method performs on par with prior work while being a bit more than 10% faster than the best performing similar method.

**Strengths:**

The proposed method is defining an interesting way to distribute information within clusters but also across clusters using attention to a set of learnable tokens.

The attention block is a crucial component in most models used today so any attempt to improve it is welcome.

**Weaknesses:**

The most important limitation of the work is the lack of sufficient comparisons with prior work. Namely, the proposed method is conceptually very similar to Luna [1] and Clustered attention [2]. Luna, similar to CAST, uses attention to learnable tokens to aggregate information and avoid computing the all to all dot product of self-attention. Although the proposed mechanism is clearly different from either of these works, it is significantly more complicated (especially compared to Luna) and it performs slightly worse albeit also slightly faster. Luna is not even discussed in the related works section (neither clustered-attention).

In addition to the above, using only the LRA benchmark to run experiments, is very limiting as it is not clear which method is better. For example, from the LRA numbers it is quite clear that structured state space models are significantly better than all other methods. Results on real world datasets however usually tell a different story where SSSMs are much closer and often worse than vanilla transformers.

Finally, a big limitation of most clustered methods is the inability to handle arbitrary masks and autoregressive causal masks which significantly limits the practical application of these models. A practical application on real data would significantly improve the argument for the proposed method.

[1]: Luna: Linear unified nested attention.
[2]: Fast transformers with clustered attention.

**Questions:**

How does the model perform without the inter-clustered attention? Performing ablations wrt the cluster size is fine however it would be significantly more interesting to ablate the components of the method like the inter cluster value.